# Evaluating the evidence trace of NeurIPS 2025 contributions with agentic code auditing

## Abstract

Code serves as the primary evidence behind computational publications, yet a detailed review of an unfamiliar codebase imposes a commonly prohibitive time burden on volunteer reviewers. Consequently, self-reported reproducibility checklists at major machine learning venues face little empirical verification, leaving code as a significant blind spot in peer review. To bridge this gap, we introduce *AuditOwl*, an autonomous, verification-centric LLM pipeline designed to make code auditing feasible for authors pre-submission and reviewers post-submission. With this framework, we conduct an audit of 100 randomly sampled empirical papers from the NeurIPS 2025 main track. For each paper, an LLM agent inspects the repository and evaluates its scientific claims against the evidence in the underlying code. Following an independent adversarial verification pass to maximize precision, the system raises 605 discrepancies (averaging 6.1 per paper). It forms a graded evidence trail: findings are quote-anchored, cite specific code locations, and about half are backed by executable verification checks that the agent implements. On a manually validated subset of findings, we measure an error rate of 6.2%. Our approach reveals a steep reproducibility funnel: only 87% of papers in our sample release code at all, and for just 9% all analyzed published findings trace cleanly to the repository. Discrepancies we find are heavily dominated by incompleteness of code and mismatches between what the paper describes and what the code does. We also detect a prevalence of technical bugs and serious methodological issues. Operating at reasonable cost, agentic code auditing can augment human peer review and help to make computational science more reproducible. Code available at: `https://github.com/anonymOwl/AuditOwl`.

## 1 Introduction

Computational science functions on the promise that the code and data released by authors are able to reproduce the headline claims in the written paper. Over the past decade, the machine learning community has acknowledged a systematic reproducibility crisis (Kapoor & Narayanan, 2023). In turn, major venues, e.g., the Neural Information Processing Systems (NeurIPS) conference, have introduced reproducibility checklists, code submission guidelines, and reproducibility challenges (Pineau et al., 2021; Sinha et al., 2022; Pineau et al., 2019). The expectation on reproducibility has been further formalized by theoretical frameworks, such as Gundersen's artifact footprint (Gundersen & Kjensmo, 2018; Gundersen, 2021), Heil's standards for archived and deterministic reproducibility (Heil et al., 2021), and the REFORMS checklist for methodological rigor (Kapoor et al., 2024). The 2019 Machine Learning Reproducibility Checklist introduced by NeurIPS (Pineau et al., 2021) aimed to raise the baseline for code and data transparency in computational science research. However, it relies on authors' self-reporting, creating a false sense of rigor that can lead to *evidential inversion*, where highly consequential claims or highly cited research are often the least reproducible (Serra-Garcia & Gneezy, 2021; Vishwarupe et al., 2026). Mere documentation of methods, such as the model architecture, hyperparameter values are insufficient in recreating the results of a paper (Heil et al., 2021). Manual recreation of models based solely on written text require a large input of time and effort, rendering them impossible to expect from volunteer reviewers (Raff, 2019).

Therefore, code is the main evidence that supports a computational paper. Many possible errors that invalidate a scientific claim, such as mismatches between what the paper claims and what the code executes, are impossible to spot from the paper alone. However, code remains the blind spot of scientific peer-review as manually searching through an unfamiliar codebase has a commonly prohibitive time cost. This leaves ample room for different classes of errors to go undetected: significant parts of the code may be *missing*, even when a GitHub or Zenodo repository exists; the *methodology* may be invalid, as with data leakage or unsound statistical tests that are not described in detail in the text; the code may contain a *bug* that can have consequences from reproducibility nuisances to invalidating claims; and, finally, the code may *mismatch* the paper, running a procedure that considerably differs from the one described (Kapoor & Narayanan, 2023; Gundersen & Kjensmo, 2018; Hutson, 2018; Haibe-Kains et al., 2020). The scientific community rests on the correctness of prior work, as they apply and refine it for new problems. When errors systematically slip through review and propagate, they lower the signal-to-noise ratio of the literature and can gradually erode trust in published results. For many conferences, post-publication corrections are very rare or only happen in extreme cases, which makes catching errors before publication especially valuable.

The surge in the number of scientific submissions in the computational field, e.g., submissions to NeurIPS increased by 15 times in the past 12 years, exacerbates the issue while causing a higher strain on reviewers' time (Baker, 2025; Tan & Liu, 2026). In 2025, over 50 peer-reviewed NeurIPS papers were found to have AI-generated citations (Ansari, 2026). We expect that an increasing share of released code is now also AI-generated, which can increase the verification burden on the reviewer.

To cope with the scale of submissions, part of the community has recently experimented with AI-assisted peer review. Large-scale deployment, such as the AAAI Conference on Artificial Intelligence 2026, used Large Language Models (LLMs) to generate reviews for nearly 23,000 papers in less than a day, and reported that authors rated AI reviews favorably over human ones on important dimensions including technical accuracy (Biswas et al., 2026). Concurrently, an expert annotation study of full peer reviews concluded that AI reviews are a valuable addition to human reviews (Kim et al., 2026). A distinct line argues for a "verification-centric" paradigm in which the AI does not imitate human reviews but instead extracts machine-readable claims for more rigorous error detection (Anonymous, 2026; Booeshaghi et al., 2026).

There are many complementary approaches to verifying computational research using agents as the catalyst. An approach is to attempt end-to-end execution, running the published code of a paper from start to finish against its data to confirm the regeneration of the same results as the numbers reported in the paper (Siegel et al., 2024; Starace et al., 2025; Yan et al., 2025). While this is the strongest form of verification, it has many bottlenecks such as the prohibitive training-time or compute costs, undocumented or costly recreation of hardware and software environments, and unreleased datasets, rendering it unfeasible in practice. Another approach is to check the manuscript against itself, extracting machine-readable claims from the text to find inconsistencies and errors within the manuscript (Bianchi et al., 2025). What is missing is a *feasible* check that bridges the gap between the claims in the paper and its code without paying the full cost of re-execution and without the ethical concerns of trying to replace a full human peer review.

In this work, we apply state-of-the-art LLM-based agents to a step that current peer review largely skips: checking that released code is evidence for the results a paper reports. Our approach is verification-centric and deliberately conservative in the scope of AI it applies. We do not aim to replace a human reviewer, but to help them to assess evidence the code provides for the scientific claims in a paper. We introduce *AuditOwl*, an open-source agentic pipeline for scientific code audits. *AuditOwl* uses Claude Opus 4.8 agents to scan the claims mentioned in a paper and inspect its released code and data repository. This shifts the paradigm to a defender's advantage, making it easy to systematically detect clear discrepancies before publication by both authors and reviewers. Specifically, we make the following contributions:

- **AuditOwl pipeline**: We open-source an agentic review system that decomposes a paper into individually testable claims and traces each claim to the code or data that should substantiate it. For every claim, it assesses whether the supporting artifact exists and can be run, whether the code matches what the paper describes, and whether the procedure the code actually implements is methodologically sound, which covers mishaps such as data leakage or underpowered baselines. Where applicable, the pipeline writes and runs its own checking scripts in an isolated sandbox to

gather concrete evidence, so verdicts rest on a reproducible check. When the code backing a claim is absent or is not runnable, the gap is recorded. Each candidate finding is anchored to a verbatim quote from the code, and re-judged by an independent adversarial verifier to minimize hallucinated defects or wrongly cited lines of code.

- **Empirical meta-audit of NeurIPS 2025**: We conduct a systematic automated audit of 100 randomly sampled empirical papers from the NeurIPS 2025 main track with manual validation on a subset.

- **Cost and feasibility analysis**: We demonstrate that agentic code audit is a highly cost-efficient augmentation for human reviewers.

Rather than judging the quality of a contribution, we identify discrepancies and reproducibility problems. Automating or augmenting full peer review carries far greater ethical implications than flagging candidate discrepancies in the code alone, so this narrower, more verifiable task is a more defensible starting point for AI assistance in peer review (Yu et al., 2024; Akella et al., 2025).

## 2 Method

### 2.1 Paper sampling and study setup

We scrape the NeurIPS 2025 *Main Conference Track* proceedings index (`https://papers.nips.cc/paper_files/paper/2025`), yielding $N = 5,286$ papers, and draw a uniform random sample without replacement. Walking the draw in order, we download the paper's pdf, classify each paper as *empirical* or *non-empirical.* We flag a paper as non-empirical only when its reproducibility-checklist entry for code/data access is `NA` and the author's justification states the code will not be shared. We require both because authors commonly interpret `NA` ambiguously. Some empirical papers select `NA` on code availability while noting, e.g., that code will be released after acceptance. For the scope of this paper, we stop at 100 empirical papers. The sample spans a broad range of subfields, from LLMs, diffusion and generative modeling, and computer vision to reinforcement learning, graph learning, causal inference, optimization and learning theory, safety and robustness, and scientific ML.

### 2.2 Pre-Audit Processing

We prepare each paper for an agentic code audit by downloading the published PDF and extracting its text with PyMuPDF. The raw extraction is normalized, e.g., by unifying line endings and rejoining line-broken words. We save the resulting text to a file, along with the PDF of the paper. The PDF can be queried by the agents for figures, tables, and exact wording, while the text file gives the agent a greppable copy with stable line anchors. Thereby, every extracted claim can be cited as a line range in the text file and re-checked by human reviewers. Furthermore, we mine candidate repository URLs from both the PDF text and its embedded hyperlink annotations, then retain only the authors' own repository as identified by author-code cue phrases (e.g. "our code is available at"), repository-name overlap with the paper title/method/acronym, or an adjacent DOI/Zenodo archive. We discard links to implementations that are only incidentally referenced, such as third-party tools. For the papers for which no code links to the main repository are found in the paper or checklist, we manually search and use deep research AI tooling to prevent false negative code availability. We shallow-clone the codebase URL, recurse submodules, and pull git-LFS content. We also fetch and unpack any source code in the paper's NeurIPS supplemental archive. The resulting self-contained folder per paper is the complete input handed to the audit agent in the next stage.

### 2.3 Agent Workflow

For the main audit, we run one autonomous agent (Claude Code[1] with Claude Opus 4.8, 1M-context) per paper, in isolation with no cross-talk between papers. Each agent receives the prepared folder and a rule-based prompt, with read-only access to the author code and a sandboxed scratch directory in which it may

---

[1] Anthropic's command-line agentic coding tool: `https://claude.com/claude-code`, v2.1.153 through v2.1.158

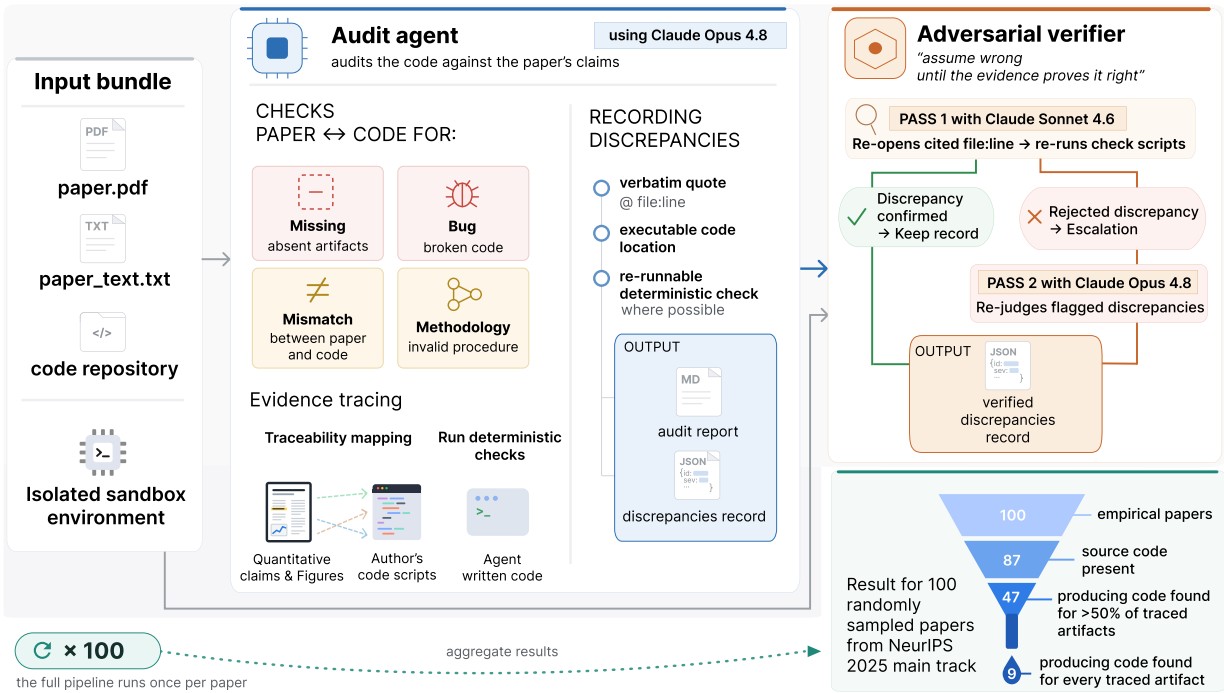

Figure 1: **Agentic reproducibility audit pipeline with *AuditOwl*.** For each of the 100 sampled papers, an input bundle is created including the paper in PDF and TXT format, as well as the cloned code repository. These along with an isolated sandbox environment are handed off to an autonomous Claude Opus 4.8 agent, which checks the paper's claims against the code for four discrepancy classes (missing artifacts, technical bugs, paper-code mismatches, and invalid methodology) by mapping quantitative claims and figures to the author's scripts and running its own tests, when necessary. Each discrepancy is recorded with a verbatim quote at file:line location and the re-executable check, when applicable, emitted as a Markdown audit report and a structured JSON machine-readable record. An independent adversarial verifier then re-checks every finding. Firstly, a Claude Sonnet 4.6 agent re-opens each cited discrepancy line and re-runs scripts in a new isolated sandbox to confirm or reject the discrepancy, while assuming the discrepancy is wrong. If the discrepancy is confirmed, the finding is stored, if not, the verifier escalates to a Claude Opus 4.8 agent to re-judge the discrepancy. Finally, all discrepancies are put together in a verified discrepancy record. The full pipeline is run for all sampled papers, and the aggregated results show a narrowing from 100 empirical papers to 87 with source code present to 9 with no missing artifacts.

write and execute its own scripts. During its reasoning loop, it reads the PDF for figures, tables, and exact wording; greps the added text file version of the paper to line-anchor each claim; explores the repository; runs deterministic checks; and emits the structured report (Figure 1).

To overcome LLM limitations in exact counting and numeric comparison and to run scripts in the author's codebase, the agent is granted execution capabilities. For deterministic checks, such as verifying train/test set overlap or evaluating statistical arithmetic, the agent writes and executes read-only Python scripts in an isolated directory that provide verifiable evidence for a claim. The agent retrieves additional external artifacts that a paper provides, such as dataset and model links. Additionally, the agent uses any data available in the code repository and is permitted to create mock data when the datasets are inaccessible to run checks. We guide the LLM agent using a rule-based system prompt structured to minimize hallucination and duplicate reporting. The prompt discourages subjective critiques of code style and mandates that all discrepancies be grounded in verifiable code references. We define discrepancies as gaps in what the paper claims and what the code can support as evidence. The audit pipeline consists of four stages:

### 2.3.1 Traceability mapping and checking claims

The main audit agent extracts quantitative claims in tables, figures, and text from the paper and maps them to the exact scripts that compute their underlying values. This generates a coverage table; an unmapped claim for which no evidence can be found in the code is a candidate discrepancy.

### 2.3.2 Discrepancy detection and categorization

Mapped claims are checked thoroughly to detect discrepancies (as outlined above). To prevent duplicate reporting and thereby overstating the finding count, the agent categorizes each discrepancy using a mutually exclusive taxonomy, and applies a single-owner rule that records each underlying discrepancy once and cross-references further occurrences instead of re-filing them as a separate discrepancy (e.g., a bug that is copied multiple times over the codebase is traced to each occurrence, but results in one recorded discrepancy).

- **Missing:** A result-producing part of the code is absent, dependencies are not specified, or the data is not available without a justification.

- **Methodology:** The code executes, but the statistical or scientific procedure is invalid (e.g., target leakage, invalid assumptions for statistical tests).

- **Bug:** The code is present but technically broken or contradicts its programmatic intent (e.g., hard-coded paths, syntax errors).

- **Mismatch:** The code is methodologically sound and runs, but deviates from the paper's description (e.g., the paper specifies a different loss function than used in the code).

The taxonomy is hierarchical in the above-listed order; a discrepancy is assigned to the highest applicable class, i.e., a mismatch caused by a bug is classified as a bug.

### 2.3.3 Audit report generation with structured evidence schema

The main audit agent's output for each paper is a single markdown report with a fixed structure: a one-paragraph summary of the audit, the coverage table, the categorized discrepancies as YAML blocks, a scorecard, and three short lists (the top 6 most consequential findings, the major components the agent actively verified as correct, and open questions for the authors). Discrepancies are output in the YAML blocks alongside a model-assigned severity, i.e., impact on the paper's conclusions if correct (high / medium / low), and confidence label, i.e., how sure the agent is that the discrepancy is correct (high / medium / low). Every block must anchor its claim to a verifiable location with a verbatim quote: for code, an exact file path and line range; for a claim grounded in the paper (e.g., missing computations for a figure or table), the PDF and a table, figure, or section reference. An extractor algorithm validates the YAML schema and extracts a machine-readable JSON with the candidate discrepancies for the agentic verification stage.

### 2.3.4 Adversarial agentic verification

Every discrepancy is then re-checked by an independent adversarial agent that sees the markdown and JSON audit results, the paper, and the repository. The verifier re-opens the cited `file:line`, confirms the verbatim quote, checks that the discrepancy's code path is reachable, re-runs any agent-written check in an isolated environment, and reasons if the cited code genuinely exhibits the claimed flaw. Verification uses a Sonnet 4.6 subagent per paper. If a verifier agent proposes to change verdicts, the cases are escalated to one independent Opus 4.8 (1M-context) subagent per paper that re-judges the discrepancies in question. All headline counts are reported after this pass.

## 3 Validation

We validate the audit on whether re-running the pipeline surfaces the same discrepancies (robustness), and what fraction of the discrepancies it raises are real and relevant issues (precision).

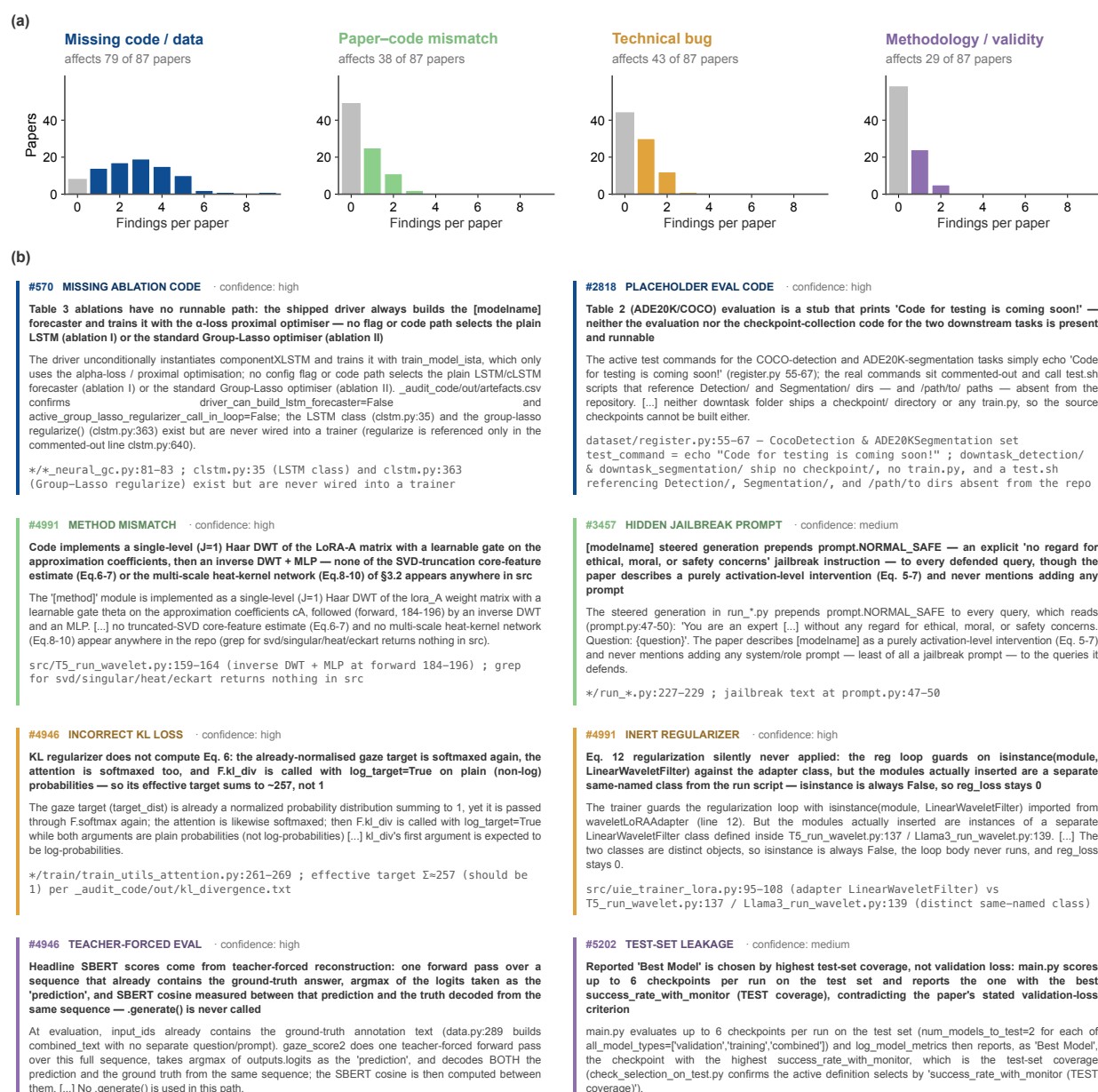

Figure 2: **Finding categories and example discrepancies. (a)** Per-paper distribution of discrepancies (excluding low-severity) across the four categories (missing code/data, paper-code mismatch, technical bug, methodology); Missingness is most common and affects 79 of 87 papers. **(b)** Two example discrepancies per category labeled with high severity as excerpts from the evidence provided by *AuditOwl* that a human reviewer can re-check. Model names or identifying paths are hidden for anonymity.

## 3.1 Robustness

We draw a uniform random sample of 5 papers from the frame of 87 audited papers that had a retrieved codebase. Each paper is audited $R = 10$ times independently (50 runs total), with model and prompt fixed. Every run executes the production pipeline end-to-end: an Opus audit pass, a Sonnet adversarial-verify pass, and Opus escalation on non-keep verdicts. Each run is a fresh subagent with no shared context, which

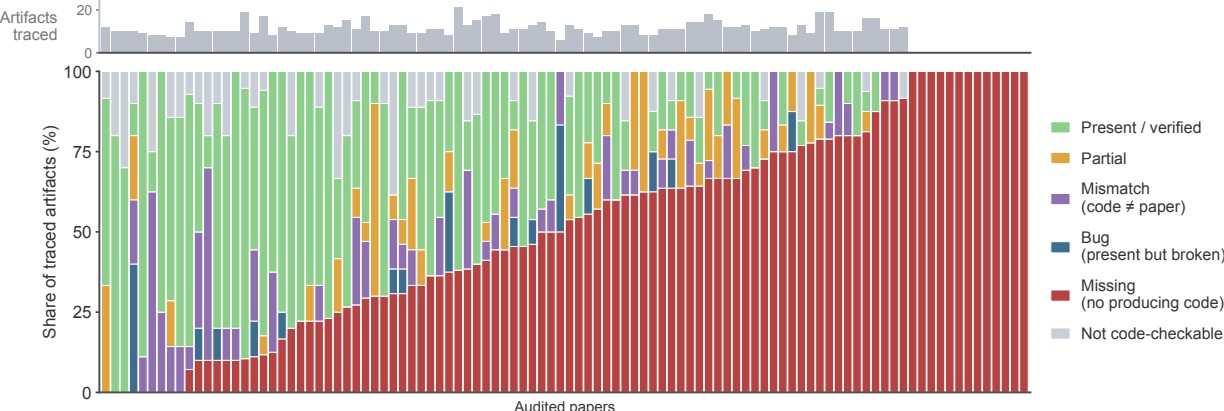

Figure 3: **Per-paper code coverage of traced results.** Each bar is one of the 100 audited papers, showing how that paper's result artifacts are backed by code evidence. An artifact is a single load-bearing result (a table, ablation, figure, or specific reported number), and the denominator is the auditor-selected set of such artifacts (skewed toward headline and suspect claims), not necessarily the paper's full figure/table count. The share of evidence status of each artifact is encoded in the bar height of the stacked bar elements. The classes are present (the producing code is in the repository; we report as present even if the data behind it is missing), partial (only partially traceable, e.g., only a part of a table can be computed or the code is incomplete), mismatch (artifact traces to code, but the code is not faithful to what is claimed in the paper), missing (the code could not be located), bug or missing inputs (code is present but does not run), and not code-checkable (e.g., human studies, or off-repository artifacts). Each artifact's status class is assigned by a deterministic keyword classifier applied to the auditor's Status cell.

never sees the prior audit. To compare findings across runs, we map the 10 runs' raw findings to distinct underlying discrepancies using an independent LLM clustering pass (Opus 4.8) that merges findings from different runs that clearly address the same underlying issues, even when formulated differently. Every raw finding is assigned to exactly one discrepancy, and the partition is checked to be complete. A discrepancy's detection rate is then the fraction of the 10 runs that surfaced it. Reliability naturally depends on discrepancy granularity (i.e., for raw findings, the detection rate is close to zero because of LLM-stochasticity); we used a single fixed prompt for our entire pipeline (not iterated against the results) and have published it on our GitHub.

### 3.2 Human validation

On the same five papers, we conduct a human validation of the discrepancies. For each paper, we merge its 10 runs' findings into distinct discrepancies (as described above, ≈16 per paper, 80 total) and manually judge every issue against the code and the paper. Each discrepancy is classified into one of five classes: correct and relevant, correct but wrong severity, correct but not relevant (i.e., the finding is a nitpick that does not influence the results or their reproducibility), unsure, or false. Thereby, we can measure the empirical precision/false positive rate.

## 4 Results

*AuditOwl* produces a re-verifiable evidence trail of code-paper discrepancies, each citing the artifact (the paper in textual form or PDF, code, supplement, or other external components) and the location in the form of a line of code or text line that justifies it, so any finding can be re-checked by an author or reviewer. We ran our agentic auditor for 100 random empirical NeurIPS 2025 main-track papers. Across the 87 papers where code could be retrieved at all, the agents checked 1,009 result artifacts and flagged 606 discrepancies, 605 of which survived the adversarial agentic re-verification. The findings come with graded verifiable evidence: All discrepancies quote the exact passage they object to, with its location in the paper, checklist, README, or

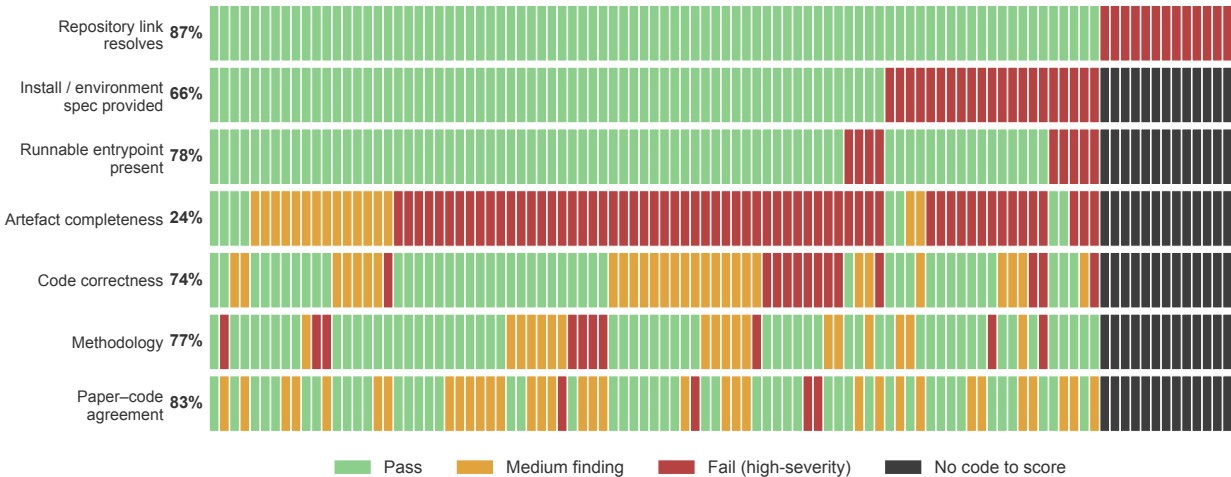

Figure 4: **Per-paper reproducibility scorecard.** Columns are the $n = 100$ audited papers; rows are eight pass/fail reproducibility criteria. The dominant failure mode is missing code: 79% of papers have at least one load-bearing result with no producing code in the released repository.

code. Further, their assessed impact on the paper's conclusion and the agent's uncertainty about the finding are indicated.

### 4.1 Discrepancy description and reproducibility scorecard

All four discrepancy categories (missing code/data, paper-code mismatches, technical bugs, erroneous methodology) are present among the 606 flagged discrepancies (Figure 2). For 86% of the 606 discrepancies, the cited evidence is a line of the authors' source code that a verifier confirmed is reachable rather than dead code; the others quote claims that exist only in the paper, such as an evaluation result for which no code line exists. For 52%, the agent additionally implemented a script that reproduces the flagged discrepancy in a sandbox. Discrepancies cover a range of agent-assessed severity and confidence levels, with overall balanced distribution of severity but skew to high auditor confidence, especially for high-severity discrepancies (Figure S1). Of note, no high severity finding is identified with low auditor confidence.

Missing code or data constitutes the most frequent category: While code was available for most papers (87/100), it was flagged as incomplete for nearly all of them (Figure 2 a). Many authors publish the core model code while omitting the surrounding evaluation, ablation, and baseline code, which makes the reported numbers hard to reproduce (see missing code examples in Figure 2 b). In a typical paper, 44% of the traced result artifacts have no corresponding code (Figure 3).

The audit also surfaces severe methodological problems that can inflate results and undermine a paper's conclusions (Figure 2). Several of them are visible only in the code. In 10 of the 87 papers that released code, *AuditOwl* found a high-severity methodological error, such as an unfair baseline comparison or leakage from the test set. Plain mismatches between what the paper claims and what the code does are common (Figure 2). Technical bugs are usually a minor reproducibility nuisance and can often be fixed with a single-line change. Rarer but more serious are silent logic errors (4 of 96 discrepancies classified as bugs) where the code runs but computes something other than the method described: *AuditOwl* finds, for example, a regularizer that is the paper's core contribution but is silently never applied (Figure 2b).

Overall, the NeurIPS guideline that "the authors should provide scripts to reproduce all experimental results for the new proposed method and baselines." (Anonymous, 2025) is rarely satisfied. We summarize discrepancy categorization and severity assessment for each paper in a reproducibility scorecard (Figure 4). Only 9 of the 100 papers substantiate every reported result *AuditOwl* mapped with code, and thus meet this guideline in full.

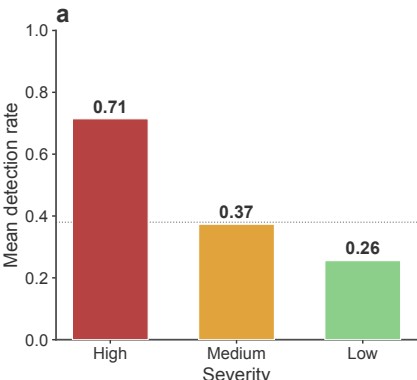
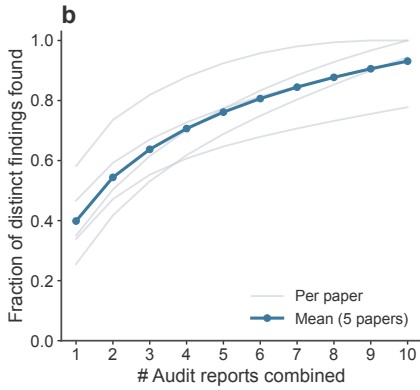

Figure 5: **Test–retest stability of the LLM reproducibility auditor.** We re-ran the full audit pipeline (Opus audit; Sonnet re-verification with Opus escalation) 10 times on each of 5 NeurIPS-2025 papers (uniform random, drawn from the 87 sampled papers that had a retrieved codebase) running every replicate in a sandbox. **(a)** Mean detection rate of discrepancies across runs stratified by severity (high $n$=14, medium $n$=30, low $n$=36). The mean detection rate is the average, over all distinct issues, of the fraction of the 10 reruns in which each issue was re-found. The dotted horizontal line represents the overall mean detection rate (0.38). Discrepancies that are classified by the agent as highly severe are more stable. **(b)** Fraction of distinct findings found when combining audit reports by merging findings that point to the same underlying issue. The saturation indicates that audit samples come from the same bounded set of real issues rather than being uncorrelated hallucinations.

## 4.2 Robustness experiments and human validation

Across 10 repeated runs on each of 5 randomly selected papers, severe discrepancies are highly reproducible, whereas medium- and low-severity issues are less consistently rediscovered (Figure 5). Repeated runs converge: pooling them keeps re-surfacing the same findings instead of a new random set each time (Figure 5b). Thus, the limited reproducibility likely reflects incomplete detection of minor issues rather than hallucinations: Under the assumption that hallucinations are only weakly correlated (if at all), we would expect a near-constant number of fresh findings per run and therefore a linear increase in Figure 5b without saturation. We conclude that the low observed detection rate is likely a sensitivity issue towards minor problems, with low-severity reproducibility nuisances often being false negatives that surface in only some runs. Ensembling can help if correcting all minor reproducibility issues is a priority.

We assess precision by human expert validation of all 80 distinct findings *AuditOwl* raised across the 5 papers in the 10 runs after merging discrepancies that point to the same issues. Overall, 74 were judged to be factually correct, 5 false, and 1 was left unsure. Human validation confirmed all but 6 severity ratings of *AuditOwl*, severity was not found understated. Sixteen issues were classified as "correct but not relevant": 14 of 16 (88%) are situational discrepancies that the human reviewer judged not to influence the reported results or reproducibility, but that also *AuditOwl* labeled as low-severity. Consistent with the robustness analysis, restricting to medium- and high-severity findings, the auditor's accuracy (correct and relevant, or correctly flagged but at an overstated severity) rises to 83% (29/35), with zero entirely false high-severity findings, versus 64% on low-severity findings. Human-rated accuracy also increases monotonically with the auditor's self-reported confidence (25% at low, 57% at medium, and 82% at high, Figure 6 b). Higher-confidence findings tend to be more correct.

Among the false findings, two were verification misses where the agent did an inaccurate search or missed domain knowledge (wrongly claiming a paper did not implement its seven-seed robustness runs, and wrongly claiming a baseline was used with non-standard hyperparameters), and three were over-interpretations of correctly-read code (methodology/validity and technical-bug claims the code implemented, but which do not manifest under the authors' protocol), all of low-to-medium severity. This analysis supports our conclusion from the robustness runs that hallucinations are not a central issue.

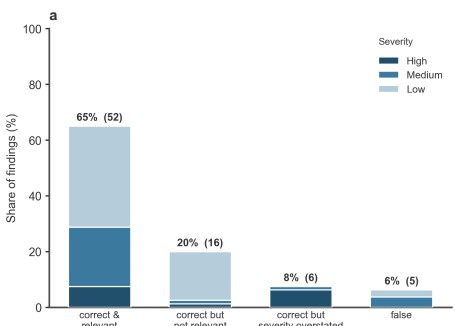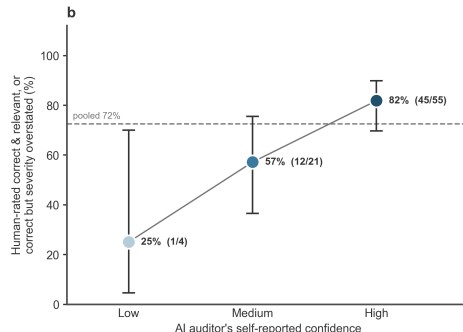

Figure 6: **Human evaluation of agentic auditors' findings. (a)** Human verdicts on all rated discrepancies (except for one finding, which was rated "unsure"). **(b)** Confidence calibration: the fraction of findings the human expert judged rated as correct and relevant (possibly with overstated severity) against the self-reported mode confidence of the agentic auditor. Error bars are Wilson 95% confidence intervals.

### 4.3 Token cost and run time

Our agentic workflow is feasible to run routinely. Across the 87 papers with retrieved code, each audit consumed a mean of 6.9M tokens, split between the main audit pass as well as the adversarial verification pass (Figure S2a). About 92% of tokens are cache reads, which most current APIs serve at a steep discount. Runtime is similarly manageable. A single paper takes a mean of 9.8 minutes to run through the audit as well as the verifier end-to-end (Figure S2b). As papers are audited in isolation with no cross-talk, we can parallelize running in batches bound only by the slowest single agent. We carried out the audits in batches of around 20 papers. The longest paper took 23.4 minutes. In total, we needed less than 100 minutes to audit all papers. Per paper, the auditor yields a median of 7 discrepancies (range between 2 and 13).

This means agentic code auditing is well within the budget of a pre-submission self-check or a reviewer-side screening, all at a fraction of the human time cost for a manual codebase audit.

## 5 Discussion

Our meta-audit on a subsample of NeurIPS 2025 papers reveals that the gap between papers' claims and their released code is large, systematic, but cheap to detect. The main/dominant discrepancy category was not methodological error but rather simple code absence (51% of the discrepancies), which does, however, make the other three discrepancy types undetectable. The average paper was missing 44% of the code backing the arguments our auditor deemed load-bearing. This was followed by paper-code mismatches which make up another 25%.

In the context of the literature on systematic reproducibility analyses, the NeurIPS papers we audit do not score well. For example, every audited paper fell short of Heil's reproducibility Heil et al. (2021), where the lowest tier (Bronze) requires code, data, and trained model weights to be deposited in an immutable, citable DOI archive (such as Zenodo) for permanent storage. This is, however, expected because NeurIPS submission guidelines do not mandate this currently.

Our numbers provide a first estimate for reproducibility in a community where these standards are currently self-reported without systematic peer review of the code. Furthermore, the discrepancy of roughly 39% of NeurIPS 2019 papers self-reporting code availability falling to around 28% once a reviewer checked code availability is evidence that author checklists may overstate the reality of code availability (Magnusson et al., 2023). Our pipeline goes beyond the trivial check of repository availability, which is a necessary but not sufficient standard, and analyzes whether the code behind the link can substantiate the claims in a paper.

The gold standard to validate a computational paper's findings is to fully reproduce the tool by re-running the released code with the data and on new datasets. However, it commonly entails prohibitive compute cost,

facing unreleased datasets, unrecoverable software environments, or external components that are hard to reproduce, and does not necessarily lead to the detection of methodological or mismatch discrepancies. Our contribution is deliberately a high-yield but low-cost audit that is complementary to the full reproduction approach. We do not attempt full reproduction, but rather audit traceability and correctness: i.e., that each headline claim maps to code capable of producing it with the same methodological steps described in the paper. Passing our code audit is a necessary condition for full reproducibility without having to re-implement the methods. This distinction is what makes our tool feasible at scale.

Running *AuditOwl* has a reasonable cost. The entire study (100 papers, 87 audits plus 50 robustness re-runs) was carried out on a single consumer Claude Max subscription rather than a metered enterprise account. Therefore, the pipeline is well-suited for deployment as a routine quality control tool. For example, authors can catch mismatches between their output writings and code, and journals or conferences can make code reviewable, without having to rely on more fully automated peer reviewing. Agentic code audit can transform code review from a difficult search problem in codebases to the validation of specific, possibly problematic line and file locations. *AuditOwl* is not designed to pass a verdict on a paper in the publishing process, but rather to assist authors and reviewers to address clear, objective issues in the code. The adversarial verification pass was designed to re-check every finding against the cited evidence to prevent hallucinated lines of code or text that would break the evidence trace. However, it only corrected 3 of 606 findings, two severity/confidence downgrades, one rejection of a false positive finding that conflated two experiments. Depending on the token budget, this step may be optional.

In our robustness experiments, when pooling independent audit reports numbers of distinct discrepancies saturates, which indicates that run-to-run variability is primarily a recall limitation rather than hallucinations: the auditor raises a different subset of true but borderline relevant discrepancies each time, instead of inventing false positives. This is confirmed by the overall low false positive rates in the human validation trial. Recall limitations of smaller issues can be simply mitigated by ensembling a small number of re-runs at a trade-off between increasing costs and recall rate. However, code can contain deeper issues that require in-depth domain knowledge, where ensembling cannot help to improve recovery.

## 6 Limitations

Agentic verification-centric code audit is an efficient method to increase reproducibility and trust in computer science publishing, however, there are limitations to our methodology. Our sample of 100 empirical papers from a single venue and year is not enough to generalize over the whole NeurIPS, let alone the machine learning publishing. The steep reproducibility funnel is, however, drastic enough for us to provide a qualitative conclusion that there is often little code evidence behind publications, even if factoring in the possible error rate of the agentic auditing.

We used Claude Opus 4.8 as the main model behind our approach, and we do not expect that our pipeline works well with much weaker models. We open-source our pipeline to enable use and performance evaluations in other settings as the model landscape behind agentic systems is rapidly evolving.

By design, we focus on auditing traceability, i.e., we locate claims and their code and check if it is implemented in a methodologically faithful way as described in the paper. We do not re-run all code to reproduce published numbers for all checks, but rather a subset, since full re-execution is unfeasible.

Reading PDFs is token-expensive and prone to loss for agents, and using automated text file creation from PDFs is not fully reliable. We input both PDFs and generated text files for the agent to consume; however, we cannot promise full input fidelity. Agentic code auditing can benefit from trends towards more machine-readable science (Booeshaghi et al., 2026). Furthermore, while we shallow-clone code repositories and pull LFS content, we do not retrieve large datasets residing outside the repository. We suspect this can create a ceiling for the recoverability of methodological problems (e.g., a simple baseline might outperform a proposed model, but this can only be tested with the dataset available).

Finally, agentic systems can make mistakes and are vulnerable to prompt injection. We imagine two classes of attack: an author might try to bias the agentic system toward a favorable audit, or try to run malicious code on the system that performs the audit. The latter can likely be mostly addressed by sandboxing; for

the former and possible mistakes, a human reviewer must validate the audit before it is used to evaluate papers and detect suspicious patterns.

A natural objection to our work here is that we only human-validate a small subset of the discrepancies which were flagged: We argue that each audit should be validated, yet we report findings across 100 papers that are not all themselves exhaustively human-validated. We, however, only characterize distributional tendencies and never judge individual work. We critique the community-level standard where code availability and completeness is self-reported but not reviewed. In line with this, our conclusions are robust to the measured error rate: human validation of the 80 examined discrepancies yielded 74 correct, with false positives concentrated in low-severity discrepancies, which we exclude in most analyses and overall reproducibility assessment. Finally, a complete human validation of discrepancies across our full benchmark presented here is not within our scope, but validating the findings for a single paper is entirely tractable for that paper's author or an expert reviewer.

## 7 Ethics Statement

Our agentic code audit system is designed to augment human review, and by no means replace it. It produces a quote-anchored human-readable evidence trail and no verdict, steering attention to specific file and line locations. The output is formatted in a way that facilitates a human expert to check every finding. Not all discrepancies are decision-relevant (this also depends on community standards and journal guidelines), and using them to pass judgment might unfairly harm authors. Our audit has flagged flaws in real published work, but examples here serve only to illustrate categories and to investigate general trends and should not be used to judge researchers, especially since the issues we observe are systematic and not individual shortcomings. We also note that missing-code rates may correlate with lab resources, so discrepancy counts measure traceability, not necessarily contribution quality, and should not be repurposed as a direct reproducibility judgment. Our tool exists to make code review feasible and scale human expertise, never to replace it.

## 8 Conclusion

Code is the primary evidence underpinning computational results, yet it remains largely unexamined during the peer-review process. We introduced *AuditOwl*, an open-source agentic pipeline that traces a paper's headline claims to its released code, writes and runs deterministic checks in a sandbox, and re-verifies all findings through a further adversarial pass. Applied to 100 randomly sampled NeurIPS 2025 papers, it produced a graded and quote-anchored evidence trail in minutes for each paper, revealing a steep reproducibility funnel, where incomplete or mismatched code is a standard occurrence as opposed to a rare finding. *AuditOwl* provides a validation-centric approach to augment, instead of automating judgment. It provides a feasible pre-submission check for authors as well as a recommendation of which parts of the code to pay attention to for reviewers. Submission and codebase volumes surge in computer science. This can outgrow available human attention needed to verify them, and tools to make human verification more efficient are needed. Making agentic paper-to-code auditing a routine step in publishing is a low-cost, yet immediately actionable pathway towards more reproducible computational science.

## 9 Reproducibility Statement

Our pipeline splits into a deterministic part and a stochastic LLM-agent-based part. The deterministic part includes randomly selecting empirical papers from the NeurIPS 2025 main track, which we do with a fixed seed. The post-processing of the audits is also fully deterministic. The audit and verify passes depend on stochastic LLMs, which we run on third-party repositories that might drift over time (we cannot redistribute the third-party code we audit). To alleviate this, we publish all discrepancies and agentic audit reports on our GitHub (`https://github.com/anonymOwl/AuditOwl`) alongside the code for pre- and post-processing and plotting. A full re-run costs about 509M tokens for the audit, 94M for the verification run, and roughly 25 minutes of time if run in parallel. Furthermore, we run *AuditOwl* for our contribution and append the resulting audit in Section S1.2. The report states that our paper's quantitative claims trace cleanly to the code. For other minor issues, we edit the paper and clean the repository pre-submission based on the results.

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

# A    Appendix

### S1.1    Supplementary Figures

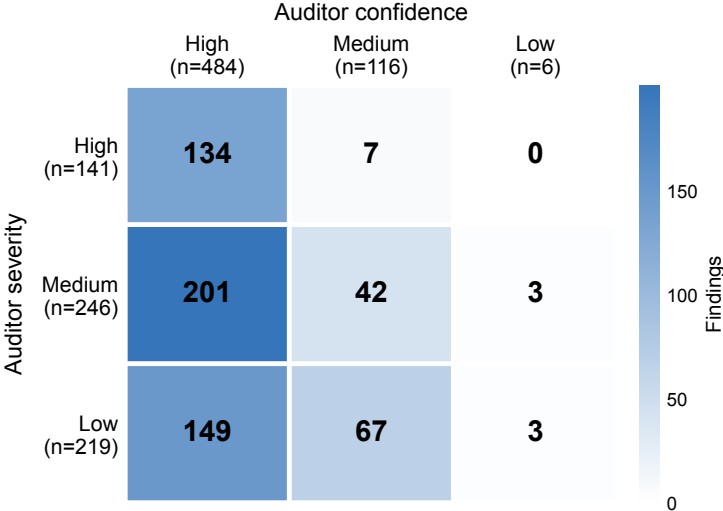

Supplementary Figure S1: **Number of flagged discrepancies by severity and confidence.** For the 606 discrepancies found in the 87 papers that released their own source code, most are flagged with medium severity at high confidence. Note both axes are the agent's own labels.

### S1.2   Supplementary Information - **Audit for this contribution**

We ran *AuditOwl* on this project, and based on the audit we received below, we made changes to the submission. We corrected the minor mismatches, or missing method descriptions, and updated the project README, and removed stale files from earlier work. According to *AuditOwl* the paper's quantitative claims reconcile cleanly against the released data. Note the resulting audit was converted into PDF from markdown and therefore has minor stylistic issues.

# Audit — AuditOwl: "Evaluating the evidence trace of NeurIPS 2025 contributions with agentic code auditing"

## Summary

This is a self ("inception") audit of the AuditOwl repository, which both implements the agentic-audit pipeline the paper describes AND ships the audit data the paper's numbers are computed from (`audits/*/findings.json`, `findings_verified.json`, `_summary/data/*.json`, `_robustness/data/*.json`, `token_cost.json`). I audited the git-tracked tree at commit `fa45b50` strictly read-only, and re-derived every headline quantity deterministically from the released data rather than trusting the prose. My checks live in `_audit_code/`: `count_findings.py` (re-totals all 87 `findings.json` + verifier verdicts → `out/count_findings.txt`) and `recompute_coverage.py` (re-imports the repo's own `build_coverage_figure.py` and re-parses every `audit.md` coverage table read-only → `out/recompute_coverage.txt`). I also re-ran the repo's own scripts where they are data-only: `random_list.py` (seed-42 sample reproduction), `print_funnel.py` (the headline funnel), and `aggregate.py` (regenerates `figure_data.json`).

**Headline verdict: the paper's quantitative claims reconcile cleanly against the released data.** I re-derived, independent of the prose, all of: 100 sampled / 87 code-present / 13 no-code; 606 raised → 605 surviving findings; the 4-way category split (missing 307, mismatch 151, bug 96, methodology 51) and the 51%/25% discussion shares; the evidence-trail fractions (99.8% quote-anchored, 86% code-located, 52% check-backed); the funnel 100→87→47→9; the "44% missing" coverage; the 1,009 traced artifacts; Figure 2a per-paper prevalence (79/38/43/29); the entire Figure S1 severity×confidence heatmap (134/7/0 · 201/42/3 · 149/67/3); the scorecard percentages (87/66/78/24/74/77/83); the human-eval error rate (5/80 = 6.25% ≈ 6.2%) with the exact 74/5/1 split; the robustness detection rates (0.71/0.37/0.26, overall 0.38, n=14/30/36); and the cost/timing (6.9M tokens/paper, 509M+94M total, 9.8 min mean, 23.4 min longest). The sampling fully reproduces: the released 5,286-row frame + seed=42 regenerates `list.csv` byte-for-byte, and all 106 audited folders map to the first 106 positions of that draw. **No headline number failed to reconcile.**

The findings below are reproducibility-hygiene and internal-consistency issues, none of which changes a conclusion: (1) two committed JSON files (`stats.json`, `funnel.json`) carry stale 50-paper numbers that contradict the paper and are read by nothing; (2) `aggregate.py`'s install/runnable/scorecard rows and its alternate funnel cannot be regenerated from the public release because they scan the git-ignored upstream `code/` trees; (3) two co-existing definitions of "all results trace cleanly" (9 vs 6 papers) — the paper uses the looser 9; (4) small paper-internal / README-vs-data wording mismatches (84% vs 86% code-located; README $13/paper vs data $14/paper).

## Traceability table

| Paper artefact | Repo location | Computed value | Matches paper | Status |
|---|---|---|---|---|
| Abstract "100 randomly sampled empirical papers" | `random_list.py:34` (seed=42) + `neurips_2025_main_track.csv` (5,286 rows) → `list.csv` | seed-42 draw reproduces `list.csv` byte-for-byte; 106 audited dirs = first 106 draws | ✓ | Verified (`_audit_code/out/count_findings.txt`) |
| "N = 5,286 papers" frame | `neurips_2025_main_track.csv` | 5,286 data rows | ✓ | Verified |

| Paper artefact | Repo location | Computed value | Matches paper | Status |
|---|---|---|---|---|
| 100 → 87 code-present → 13 no-code | `audits/[0-9]*/findings.json` presence | 100 dirs, 87 with findings.json, 13 without | ✓ | Verified |
| "605 discrepancies … 605 survived" | `audits/*/findings_verified.json` (keep+lowered) | 606 gross, 605 survive (603 keep, 2 lower, 1 reject) | ✓ | Verified |
| "averaging 6.1 per paper" | 605/100 = 6.05 | 6.05 (≈6.1) | ≈ | Verified (rounding) |
| "median 7 (range 2–13)" | per-paper finding counts | median 7, min 2, max 13 | ✓ | Verified |
| Category split (Fig 2 / S1) | `figure_data.json:by_category` | missing 307, diff 151, bug 96, meth 51 | ✓ | Verified |
| "missing 51%", "mismatch 25%" (Discussion) | 307/605, 151/605 | 50.7%, 25.0% | ✓ | Verified |
| "1,009 result artifacts" | `coverage_status.json:n_artefacts` (parsed from audit.md tables) | 1,009 | ✓ | Verified (`recompute_coverage.txt`) |
| Funnel 100→87→47→9 (Fig 1) | `print_funnel.py` ← `coverage_status.json` | 100, 87, 47, 9 | ✓ | Verified |
| "44% of traced artifacts missing" (Fig 3 / Disc.) | `coverage_status.json` mean/median missing_frac | mean 0.434 / median 0.444 | ✓ | Verified |
| "9 of 100 substantiate every result" | `coverage_status.json` missing_frac==0 | 9 | ✓ | Verified |
| Evidence: 99.8% quote-anchored | `aggregate.py:546` n_quote/N | 605/606 = 99.8% | ✓ | Verified |
| Evidence: "84%"/"86%" code-located | `aggregate.py:547` evidence_kind=="code" | 519/606 = 85.6% | ✗ (paper §4.1 says 84%) | MISMATCH → finding `code-located-pct-84-vs-86` |
| Evidence: "52%" check-backed | `aggregate.py:548` n_check/N | 315/606 = 52.0% | ✓ | Verified |
| Fig 2a prevalence 79/38/43/29 (excl. low sev) | per-paper surviving findings by category | 79, 38, 43, 29 of 87 | ✓ | Verified |
| Fig S1 heatmap cells | `audits/*/findings.json` sev×conf (gross) | high[134,7,0] med[201,42,3] low[149,67,3] | ✓ | Verified |
| "no high-sev finding at low confidence" | sev=high ∧ conf=low | 0 | ✓ | Verified |
| Fig 4 scorecard 87/66/78/24/74/77/83 | `figure_data.json:scorecard_with_med` | 87/66/78/24/74/77/83 | ✓ | Verified (values frozen; see finding `aggregate-funnel-needs-codetree`) |

| Paper artefact | Repo location | Computed value | Matches paper | Status |
|---|---|---|---|---|
| "10 of 87 high-sev methodology" | `figure_data.json:severity_breakdown:high_papers:methodology` | 10 | ✓ | Verified |
| Verifier "corrected 3 of 606" | `findings_verified.json` verdicts | 2 lowered + 1 reject = 3 | ✓ | Verified |
| Human-eval "error rate 6.2%" | `human_eval_summary.json:pooled` | 5 false / 80 = 6.25% | ✓ | Verified |
| Human-eval 74 correct / 5 false / 1 unsure | `human_eval_summary.json:pooled` | 74 / 5 / 1 | ✓ | Verified |
| Robustness detection 0.71/0.37/0.26 (Fig 5a) | `robustness_figure_data.json:by_severity_majority:merged` | 0.714/0.373/0.256, n=14/30/36 | ✓ | Verified |
| Robustness overall mean 0.38 | `robustness_figure_data.json:overall_mean_detection:merged` | 0.38 | ✓ | Verified |
| Cost 6.9M tokens/paper (Fig S2a) | `compute_cost.json` audit+verify per code paper | 6.927M | ✓ | Verified |
| 509M audit + 94M verify tokens (Repro. Stmt) | `compute_cost.json:totals` | 509.1M + 93.5M | ✓ | Verified |
| "≈92% cache reads" | per-paper `token_cost.json` cache_read share | 93.4% | ≈ | Verified (rounding) |
| 9.8 min mean, 23.4 min longest (Fig S2b) | `compute_cost.json` wall_min | 9.81 mean, 23.42 max | ✓ | Verified |
| README "$13/paper" (NOT in paper body) | `compute_cost.json` audit_cost_usd | mean $14.09 / median $12.73 / total $1,409 | ✗ (README says $13.32/$12.16/$1,332) | MISMATCH → finding `readme-cost-vs-data` |
| Committed `stats.json` / `funnel.json` | `_summary/data/stats.json`, `funnel.json` | 50 papers / 323 findings / funnel total 50 | ✗ (contradict the 100/606 paper) | STALE → finding `stale-stats-funnel-json` |

## missing

(No `missing` finding rises to a real defect. The paper's "Supplementary Information — Audit for this contribution" is marked "TBA" in the released text — see `paper.pdf` §S1.2 — but that is an acknowledged placeholder in a work under review, explicitly excluded by Rule "ignore minor issues the authors have already acknowledged", and is not a result-producing artefact. Reported as an open question below rather than a finding.)

## bug

``yaml finding id: aggregate–funnel–needs–codetree category: bug topic: "result traceability / reproducibility scope" title: "aggregate.py funnel + scorecard rows can't regenerate from the public release (need git–ignored code/ trees)" severity: medium confidence: high status: finding file: _summary/aggregate.py line_start: 480 line_end: 484 quote: | has_code = len(repos) > 0 and prov_core != "no" cl = d / "code_links.txt" url_ok = cl.exists() and bool(cl.read_text().strip()) inst_ok = any(repo_has_install(r) for r in repos) run_ok = any(repo_has_runnable(r) for r in repos) claim: "aggregate.py derives has_code / inst_ok / run_ok (and thus its funnel stages 'Source code present'..'Runnable entrypoint' and the scorecard install/runnable rows) by scanning per–paper code/<owner>__<repo>/ trees, but those upstream trees are git–ignored (.gitignore:16**/code/) and absent from the released repo, so repos is empty for every paper." concern: "Re–running python _summary/aggregate.py` on the public release collapses the funnel to 100→0→0→0→0→0 and cannot reproduce the committed figure_data.json funnel [100,87,66,62,45,3] or the Fig 4 install/runnable scorecard percentages (66%/78%); those figure numbers are frozen in figure_data.json and not independently reproducible from released artefacts." resolution: "Either commit the per-paper repo fingerprints aggregate.py needs (install-file/runnable booleans, code-file counts) as data so the funnel/scorecard regenerate offline, or document in the README that aggregate.py's environment-readiness rows require the upstream code/ trees while the coverage funnel (print_funnel.py) and finding counts are release-reproducible." cross_refs: ["stale-stats-funnel-json", "trace-clean-9-vs-6"] check_script: _audit_code/recompute_coverage.py paper_ref: "Figure 1 funnel; Figure 4 scorecard rows 'Install/environment' and 'Runnable entrypoint'" validator_pass: quote_match: true control_flow: true condition_satisfiable: true

```
## difference

```yaml finding
id: stale-stats-funnel-json
category: difference
topic: "reproducibility hygiene / stale artefacts"
title: "Committed stats.json / funnel.json hold stale 50-paper numbers that contradict the
paper"
severity: low
confidence: high
status: finding
file: _summary/data/stats.json
line_start: 2
line_end: 3
quote: |
  "n_papers": 50,
  "n_findings": 323,
claim: "_summary/data/stats.json (n_papers=50, n_findings=323) and _summary/data/funnel.json
(stage counts 50/42/24/20/4, total 50) are committed but describe an earlier 50-paper run;
grep finds no .py that reads OR writes either file, so they feed no figure or headline
number."
concern: "A reader inspecting _summary/data/ sees two top-level summary files asserting 50
papers / 323 findings, flatly contradicting the paper's 100 papers / 606 findings, with
nothing flagging them as superseded."
resolution: "Delete the orphaned stats.json/funnel.json or regenerate them from the current
100-paper audit set (the live equivalents are figure_data.json and print_funnel.py's
coverage_status.json, both of which carry the correct numbers)."
cross_refs: ["aggregate-funnel-needs-codetree"]
check_script: _audit_code/count_findings.py
paper_ref: "Abstract '605 discrepancies'; Figure 1 funnel"
validator_pass:
  quote_match: true
  control_flow: true
  condition_satisfiable: true
```

```yaml finding id: trace-clean-9-vs-6 category: difference topic: "result traceability / definition" title: "Two co-existing 'all results trace cleanly' counts (9 vs 6 papers); paper uses the looser 9" severity: low confidence: high status: finding file: _summary/print_funnel.py line_start: 28 line_end: 29 quote: | (sum(p["missing_frac"] < 0.5 for p in real), "producing code found for >50% of traced artifacts"), (sum(p["missing_frac"] == 0.0 for p in real), "producing code found for every traced artifact"), claim: "print_funnel.py defines 'every result traces' as a paper whose audit.md coverage table has zero MISSING rows (missing_frac==0.0) → 9 papers (the paper's Fig 1 / abstract '9%'); aggregate.py and _summary/SUMMARY.md define it via a topic-anchored surviving-finding gate (no 'result traceability' missing finding) → 6 papers (figure_data.json funnel stage = 3 once also gated on install+runnable; SUMMARY reports 6/87)." concern: "The same English phrase ('all published results trace cleanly') maps to two different numbers in the repo (9 by coverage table, 6 by finding topic), and the abstract/Figure 1 silently use the more favourable 9 while the README headline uses 6; a reader cannot tell which definition produced the headline without reading both scripts." resolution: "State in the paper which operationalization '9%' uses (coverage-table missing_frac==0 over 87 source-present papers) and note the stricter topic-anchored count is 6/87, so the two are not conflated." cross_refs: ["aggregate-funnel-needs-codetree"] check_script: _audit_code/recompute_coverage.py paper_ref: "Abstract 'for just 9% all analyzed published findings trace cleanly'; Figure 1 ('9')" validator_pass: quote_match: true control_flow: true condition_satisfiable: true

```yaml finding
id: code-located-pct-84-vs-86
category: difference
topic: "evidence-trail reporting"
title: "Paper §4.1 says 84% of findings cite source code; the repo's own metric is 86%
(519/606)"
severity: low
confidence: high
status: finding
file: paper.pdf
quote: |
  For 84% of the 606 discrepancies, the cited evidence is a line of the authors' source code
that a verifier confirmed is reachable rather than dead code
claim: "The aggregate.py evidence_kind=='code' share is 519/606 = 85.6% (README rounds it to
86%, README.md:12). The paper §4.1 reports 84%, evidently a stricter subset — code-located
AND verifier-confirmed-reachable (code ∧ validator_pass.control_flow=true = 501/606 = 82.7%)
— but the text does not state the stricter denominator, so the same quantity reads as 84% in
the paper and 86% in the README."
concern: "An internal inconsistency (84% in the paper, 86% in the README) on the same
evidence-trail statistic, with the paper's stricter '84%' definition unstated; minor but it
is a headline evidence-quality number."
resolution: "Reconcile the paper's 84% and the README's 86%: either report the same
evidence_kind=='code' figure (86%) in both, or define the 'verifier-confirmed-reachable'
subset (code ∧ control_flow) and cite the script that computes it."
cross_refs: []
check_script: _audit_code/count_findings.py
paper_ref: "Section 4.1, '84% of the 606 discrepancies'"
validator_pass:
  quote_match: true
  control_flow: true
  condition_satisfiable: true
```

```yaml finding id: readme-cost-vs-data category: difference topic: "cost accounting" title: "README per-paper cost ($13.32/$12.16/$1,332) disagrees with committed compute_cost.json ($14.09/$12.73/$1,409)" severity: low confidence: high status: finding file: README.md line_start: 256 line_end: 257 quote: | - ~\$13 per paper for the audit pass (median \$12.16, mean \$13.32, \$1,332 total over all 100 papers; the two-stage verification adds ~\$2/paper, \$214 total) claim: "README quotes mean $13.32 / median $12.16 / $1,332 total audit cost over 100 papers, but _summary/data/compute_cost.json (the committed source) totals audit_cost_usd = $1,409.09 (mean $14.09, median $12.73 over 100 papers), and the verify total is $232.69 not $214." concern: "The README cost figures predate or diverge from the committed compute_cost.json by ~$77 (audit) / ~$19 (verify); the paper body itself states no per-paper dollar amount (only 'reasonable cost' +

token counts, which DO reconcile), so no paper claim is affected, but the repo's two cost statements disagree." resolution: "Regenerate the README cost line from the current compute_cost.json (or note the pricing/date the README figures used); the paper's token-based cost claims (6.9M/paper, 509M+94M total) are unaffected and reconcile exactly." cross_refs: [] check_script: _audit_code/count_findings.py paper_ref: "Section 4.3 'reasonable cost' (paper states no $ figure); README cost profile" validator_pass: quote_match: true control_flow: true condition_satisfiable: true

```yaml
## methodology

```yaml finding
id: coverage-heuristic-text-classifier
category: methodology
topic: "result traceability / denominator construction"
title: "Headline coverage numbers (1009 artifacts, 44% missing, funnel 9) rest on a regex
classifier over free-text Status cells"
severity: low
confidence: high
status: finding
file: _summary/build_coverage_figure.py
line_start: 95
line_end: 99
quote: |
  def classify(art: str, raw: str) -> str | None:
      """Map a (artefact, Status) row to one of six outcome buckets.

      The Status cells are free text written by the auditor, so this is heuristic.
      Ordering is load-bearing:
claim: "The 1,009-artifact denominator, the 44%-missing rate, and the 9-papers-clean funnel
are all produced by parsing each audit.md coverage table and running classify() - a hand-
tuned cascade of substring rules ('leads_missing', RUN_FAIL keywords, NA_CUES) over Status
cells the LLM auditor wrote in free text; the auditor also self-selected which artefacts to
trace (the denominator is explicitly auditor-selected, skewed to suspect claims, per the
script docstring and Fig 3 caption)."
concern: "Two layers of subjectivity sit under the headline coverage funnel - auditor-chosen
denominators and a heuristic text classifier - so the precise 44%/9 values are sensitive to
wording and rule order; the paper does acknowledge the auditor-selected denominator (Fig 3
caption) but not the classifier's heuristic mapping."
resolution: "Report a robustness check on classify() (e.g. hand-label a sample of Status
cells and give the classifier's agreement), or have the auditor emit a structured status
enum per artefact instead of free text, so the coverage counts do not depend on regex over
prose."
cross_refs: ["trace-clean-9-vs-6"]
check_script: _audit_code/recompute_coverage.py
paper_ref: "Figure 3 caption ('the denominator is the auditor-selected set'); Discussion
'44%'"
validator_pass:
  quote_match: true
  control_flow: true
  condition_satisfiable: true
```

## Scoreboard

| Category | # findings | Max severity | Note (one line) |
|---|---|---|---|
| missing | 0 | - | No missing result-producing artefact; S1.2 "TBA" is an acknowledged placeholder (open question). |
| bug | 1 | medium | aggregate.py funnel/scorecard rows can't regenerate from the public release (need git-ignored code/). |
| difference | 4 | low | Stale stats/funnel.json; 9-vs-6 clean-trace defs; 84%-vs-86%; README-vs-data cost. |
| methodology | 1 | low | Coverage headline numbers rest on a heuristic text classifier over auditor-written Status cells. |

## Top take-aways (≤6)

1. **Every headline number reconciles against the released data.** [all categories] I re-derived, independently of the prose, 605/606 findings, the category split, the evidence-trail fractions, the funnel (100→87→47→9), "44% missing", 1,009 artifacts, the full S1 heatmap, the scorecard percentages, the 6.2% human-eval error rate, the robustness detection rates, and the cost/timing — all match. The seed-42 sample reproduces byte-for-byte.

2. `aggregate.py`**'s funnel + install/runnable scorecard rows are not release-reproducible.** [bug] They scan the git-ignored upstream `code/` trees; re-running on the public repo collapses the funnel to 87→0. Those figure values are frozen in `figure_data.json`. The paper's load-bearing funnel ("9") comes from the data-only `print_funnel.py` and DOES reproduce.

3. **Two co-existing definitions of "all results trace cleanly" (9 vs 6 papers); the paper uses the looser 9.** [difference] The abstract/Fig 1 "9%" is the coverage-table count; the README/SUMMARY headline is the stricter topic-anchored 6/87. Both trace to code, but the paper should state which operationalization "9%" uses.

4. **Stale `stats.json` / `funnel.json` contradict the paper (50 papers / 323 findings).** [difference] Orphaned — read and written by nothing — but they sit in `_summary/data/` asserting the wrong totals.

5. **The headline coverage numbers rest on a heuristic regex classifier over free-text Status cells** the LLM auditor wrote, plus auditor-selected denominators. [methodology] The auditor-selected denominator is disclosed (Fig 3 caption); the classifier heuristic is not, and a robustness check on it would strengthen the coverage claims.

6. **Minor paper-internal / README-vs-data wording mismatches.** [difference] §4.1 "84%" vs README "86%" code-located; README "$13/paper" vs data "$14/paper" (the paper body states no $ figure, so its actual cost claims are unaffected).

## Items that genuinely look fine

- **Sampling is fully reproducible and honest.** Released 5,286-row frame + `seed=42` regenerates `list.csv` exactly; all 106 audited folders are the first 106 draws, consistent with the "walk the draw, screen for empirical, backfill" procedure the paper and `SAMPLING.md` describe (`_audit_code/out/count_findings.txt`).

- **The adversarial verifier is real and matches the paper.** `adversarial_verifier_prompt.md` implements the "assume-wrong-until-proven", re-open-file:line, re-run-check stance; verdicts (603 keep / 2 lower / 1 reject) reconcile with "corrected 3 of 606", and `findings_verified.json` exists for all 87 code papers.

- `aggregate.py`**'s epistemic design is sound:** figure numbers are computed from structured fields and filesystem checks, never from the LLM severity label; the `is_dropped()` correction layer (verifier reject + supplement FP + provenance FP) is applied consistently to the reproducibility-reality counts while the verification figures report gross verdicts — and re-running it reproduced `figure_data.json` exactly except for the code-tree-dependent funnel.

- **Human-eval and robustness data are complete and self-consistent**: per-paper + pooled human-eval JSON sum to the reported 74/5/1; the robustness severity strata (n=14/30/36) and the saturation curve match Figure 5.

- **Cost/timing claims reconcile to the token-level data** (6.9M/paper, 509M+94M, 9.8/23.4 min, ~93% cache reads).

## Open questions for the authors

- **S1.2 "Supplementary Information — Audit for this contribution" is "TBA"** in the released text (`paper.pdf` §S1.2). Will the camera-ready include AuditOwl's self-audit, and is it the one in this folder? (Not filed as a finding: acknowledged placeholder in a paper under review.)
- **Which exact denominator gives "averaging 6.1 per paper"?** 605/100 = 6.05 and 606/100 = 6.06 both round to ~6.1 only generously; 605/87 = 6.95 ≈ the README's 7.0. Confirm the abstract's "6.1" is surviving-findings ÷ 100-sampled. (Low-severity; the median "7 (2–13)" reconciles exactly.)
- **Will `stats.json` / `funnel.json` be removed or regenerated**, given they currently assert 50-paper totals in the published data directory?

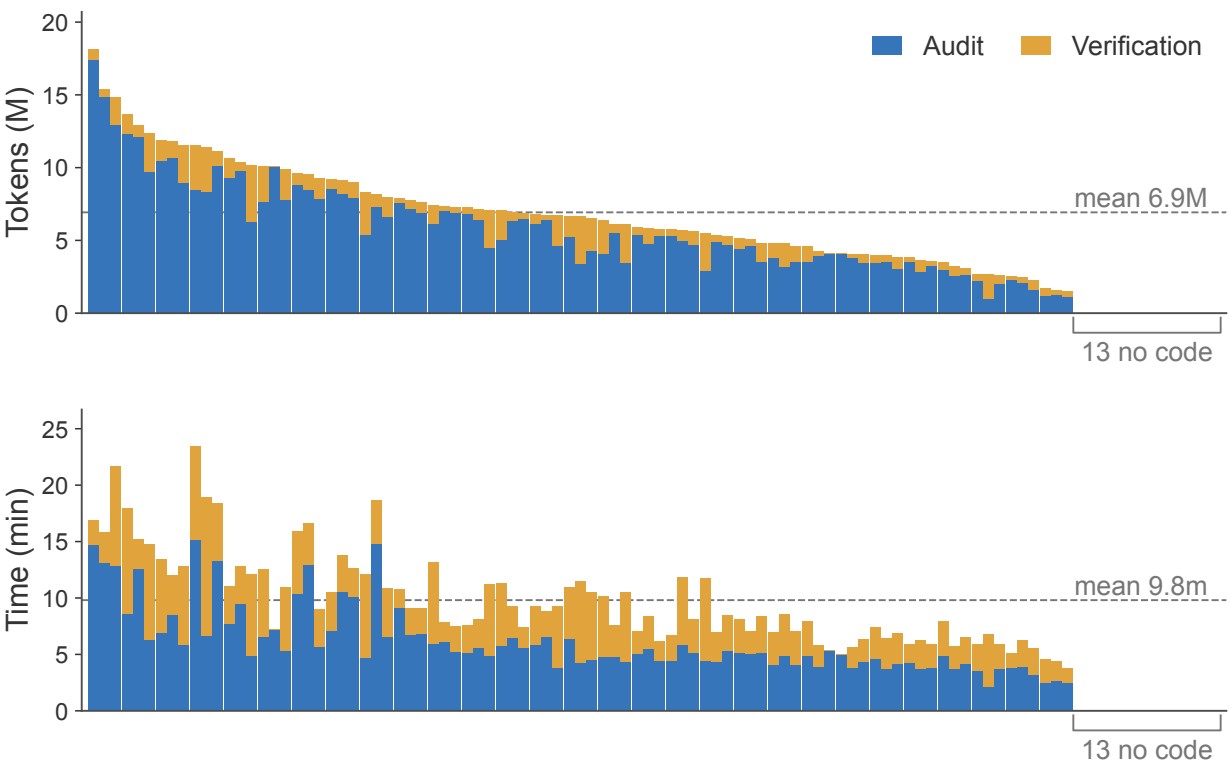

Supplementary Figure S2: **Effort per audit.** Per-paper cost and runtime for 100 papers (87 run audits, as 13 had no retrievable code). **(a)** Tokens per paper, split into the audit pass (blue), and the adversarial verification pass (orange); the mean is 6.9M tokens per paper. Across the full batch, total consumption is around 603M tokens, where approximately 94M are the agentic verifier. **(b)** Wall-clock time per paper, mean 9.8 min end-to-end (including Opus audit, Sonnet verification, Opus escalation). As the agents run in isolation, audits can run cleanly in parallel and only take as long as the longest paper requires. In our papers, the longest audit was approximately 23.4 minutes.

