# OpenReview forum: "Evaluating the evidence trace of NeurIPS 2025 contributions with agentic code auditing"
_TMLR — Under review for TMLR_

### Review · Reviewer_gmmn · 2026-07-02

**Summary Of Contributions:**

This paper introduces AuditOwl, an LLM-agent pipeline that checks whether a computational paper's released code actually substantiates the claims made in its text. Rather than attempting full re-execution (too costly) or imitating a human review, the authors stake out a narrower, "verification-centric" task: trace each headline claim to the code that should produce it, and flag discrepancies. The pipeline decomposes a paper into testable claims, maps them to specific scripts, categorizes any gaps into a hierarchical taxonomy, writes and runs deterministic checks in a sandbox where possible, and anchors every finding to a verbatim file:line quote. A separate "adversarial" verification pass re-checks each finding under an assume-it's-wrong stance.

## Key strengths

- **Well-scoped, well-motivated problem framing.** The decision to audit traceability rather than attempt full reproduction or pass quality verdicts is thoughtfully argued. Code genuinely is a blind spot in review and is important to address.
- **Falsifiable, re-checkable output.** Quote-anchored findings with file:line locations enables a human to verify any single finding cheaply, which is exactly what makes the tool usable in practice.
- **Great transparency.** The "inception" self-audit is genuinely commendable. Running the tool on its own repository is a level of intellectual honesty rarely seen in method papers.
- **Feasibility is convincingly demonstrated.** The cost/runtime numbers are concrete and low enough to make the "routine pre-submission check" pitch credible.
- **Open-sourced pipeline and data**, with a deterministic/stochastic split clearly delineated.

## Key weaknesses

- **Recall is essentially unmeasured, which undermines the main claim.** While the method shows the capability to verify its detected code problem, it does not necessarily give a good understanding on how likely the problem will be detected if it exist and if all problems in the paper can be detected, which may weaken the utility of the work as a tool of review.
- **The human validation is thin and its independence is unclear.** 80 findings on 5 papers, with no reported inter-rater reliability, no blinding, and no description of validator expertise. Given the sample spans LLMs, diffusion, RL, causal inference, theory, etc., a single validator cannot plausibly hold domain expertise across all of it — yet judging "methodology" and "test-set leakage" findings requires exactly that. The headline 6.2% error rate rests on this narrow, possibly non-independent basis.
- **Evaluator and generator share a model family.** Auditor, verifier, clustering, and escalation are all Claude models. The adversarial pass overturned only **3 of 606** findings, which the authors present as evidence of precision — but it is equally consistent with a verifier that shares the auditor's blind spots. Notably, the human check found ~6% false positives that the "adversarial" verifier let through, a tension the paper doesn't fully reconcile. A cross-family verifier would have strengthened the argument considerably.
- **Reproducibility of the study itself is fragile.** Results depend on a proprietary, drifting model, and the authors state the pipeline won't work with weaker ones. This is honestly acknowledged but is a real limitation for a paper whose subject is reproducibility.

**Audience:**

Yes

**Audience Explanation:**

This paper provides an interesting method to audit open source code accompanying paper submissions. From the methodology perspective, the proposed agentic framework is of interest for people working on automatic research and/or coding agent. From broader impact perspective, all TMLR audiences could benefit from the product of this paper.

**Broader Impact Concerns:**

No broader impact concerns.

**Claims And Evidence:**

Yes

**Claims Explanation:**

The paper convincingly shows that a cheap agentic auditor can surface many real, re-checkable code-paper discrepancies, and it documents that claim with great transparency and well-documented experimental measures. Meanwhile, the ultimate claim that the tool improves human review still need more evidence, because recall is unmeasured, the human validation is small and not clearly independent, and no human-baseline comparison exists.

**Requested Changes:**

## Critical to securing my recommendation for acceptance

These address places where a headline claim currently outruns its evidence. I'd want them resolved before acceptance, though most are reframing/analysis rather than large new experiments.

**1. Quantify or explicitly bound recall.** Precision is measured; recall is not, yet the 9%-clean figure is a recall-sensitive claim (a "clean" paper may just have had its issues missed). At minimum, seed a set of known-planted or previously-documented defects into a handful of repositories and report the detection rate, or use the ensembling saturation curve to give a defensible recall estimate with stated assumptions. Without some anchor on recall, the "clean" fraction can't be interpreted.

**2. Strengthen the human validation's credibility.** The 6.2% error rate — the paper's core precision claim — rests on 80 findings, 5 papers, with no reported inter-rater agreement, no blinding, and no statement of validator domain expertise. I'd like to see: more than one annotator on at least a subset, a reported agreement statistic, a description of who validated and how domain expertise was handled for methodology/leakage findings, and ideally blinding to the agent's severity/confidence labels. This need not be huge, but as currently reported it's too thin to carry the headline number.

**3. Address the shared-model-family concern for the verifier.** I'd like either a cross-family verifier run on a subset (to show the low overturn rate isn't shared blind spots) or an explicit, prominent caveat that the adversarial pass does not provide independent precision evidence and that the human validation is the only independent check.

## Would strengthen the work (not required for my recommendation)

**4. A human-baseline or human-in-the-loop comparison.** The paper's stated purpose is augmenting review, but no experiment shows a reviewer does better/faster with the tool, or that it catches things an expert auditing the same code would miss. Even a small study (e.g., experts audit N codebases unaided, then see AuditOwl's findings) would move the utility claim from plausible to demonstrated. I'm not making this critical because the paper is honestly scoped as a measurement/feasibility contribution rather than a utility study — but it's the single most valuable addition.

---

### Review · Reviewer_CPDr · 2026-07-14

**Summary Of Contributions:**

The paper introduces AuditOwl, an LLM-based system that checks whether a paper’s released code supports its empirical claims. It maps reported results to code, identifies missing artifacts, bugs, methodological issues, and paper–code mismatches, and verifies findings with a second agent. Applied to 100 randomly selected NeurIPS 2025 papers, it finds code for 87 papers and reports 605 verified discrepancies across 1,009 traced results. Missing evaluation, ablation, and baseline code is the most common problem, and only nine papers have code supporting every audited artifact. Human validation suggests that most findings are factually correct, particularly high-severity ones, though the evaluation covers only five papers. Overall, the paper argues that agentic code auditing can provide a practical and inexpensive supplement to human peer review.

**Additional Comments:**

The paper studies an important problem in the community. While the proposed method may not be perfect, it can be served as a reference to the researchers. Nevertheless, a more developed pipeline is expected and would benefit the community more.

**Audience:**

Yes

**Audience Explanation:**

- The paper studies an important and well-motivated problem. It targets a genuine blind spot in peer review: checking whether released code supports the claims in the manuscript is valuable but usually too time-consuming for reviewers.
- Rather than producing an ungrounded general review, the proposed AuditOwl provides file-and-line-level evidence and  executable verification scripts. This makes its outputs substantially easier for authors and reviewers to inspect than conventional LLM-generated reviews.

**Broader Impact Concerns:**

None.

**Claims And Evidence:**

No

**Claims Explanation:**

- While I appreciate the effort, only 80 findings from five papers were manually assessed, and the same five papers were used for both robustness analysis and precision evaluation. Therefore, it may not be representative enough and the conclusion requires more uncertainty analysis.
- The main auditor, discrepancy clustering, and escalated verification all rely on Claude models, with no comparison against alternative models, simpler code-analysis baselines, or human reviewers under a comparable time budget. A finer study with different model family is needed to exclude the influence of the auditor model.

**Requested Changes:**

- It would be great if the authors can elaborate how the human validation is conducted. For instance, how many annotators evaluated each discrepancy, what expertise did they have, were they blinded to the agent’s severity and confidence labels, and was inter-annotator agreement measured? The quality of the human feedback needs to be assessed, as the human decisions can be subjective.

---

### Review · Reviewer_7APT · 2026-07-20

**Summary Of Contributions:**

The paper introduces AuditOwl, an agentic system that links quantitative claims in ML papers to the code intended to support them, flags missing artifacts, bugs, methodological problems, and paper–code mismatches, and records each finding with code locations and, where possible, executable checks. A second adversarial agent re-verifies the findings. It applies the system to 100 randomly sampled empirical NeurIPS 2025 papers, reporting 605 verified discrepancies across the 87 papers with retrievable code and finding that only nine papers had code support for every audited result artifact. It also evaluates run-to-run stability, manually assesses 80 findings from five papers, and analyzes runtime and computational cost.

The paper offers a clear system with potentially high reviewer utility and seems to involve significant engineering effort, but I believe it suffers from certain methodological flaws.

**Audience:**

Yes

**Audience Explanation:**

The proposed framework can potentially offer high reviewer utility, and for this reason can be relevant to general audiences, even outside of agentic AI and LLMs.

**Broader Impact Concerns:**

The authors include an Ethics Statement Section in their work.

**Claims And Evidence:**

No

**Claims Explanation:**

Strengths
- Well-motivated paper that is potentially of high utility to reviewers.
- The experimental design is strong: quote anchoring, exact file locations, structured records, executable checks, and explicit severity/confidence labels.
- Extensive engineering pipeline that handles PDFs, repository retrieval, claim mapping, code execution, report generation, and verification.
- Good empirical examples, e.g., missing ablations, train-test leakage or paper-code mismatches. This can be a very useful tool t reviewers.

Weaknesses
- The adversarial verified (Sonnet) comes from the same family as Opus. The fact that verification changed only three of 606 findings could mean the initial auditor is excellent, but it could equally mean that the verifier shares its errors.
- The validation appears to involve a single “human expert,” without reporting evaluator expertise, independent ratings, inter-rater agreement etc. The same five papers are used for both robustness analysis and human validation. Five papers are too few to establish performance across diverse ML/AI domains.
- The mean rediscovery rate is only 0.38, with rates of 0.71, 0.37, and 0.26 for high-, medium-, and low-severity issues. This demonstrates considerable run-to-run instability. Moreover, we only consider issues discovered by the model across ten runs. But we lack an exhaustive  list of the codebase’s problems by a human. Consequently, the experiment cannot estimate recall against ground truth. It only measures self-rediscovery within the system’s own discovered set.
- The authors explicitly assume hallucinations are weakly correlated, but do not test that assumption.
- The paper emphasizes the total number of detected discrepancies, but many findings are minor or ultimately irrelevant to the validity and reproducibility of the main results. In a reviewer-facing setting, surfacing such issues may increase rather than reduce the review burden by diverting limited attention from consequential errors. The evaluation should therefore focus more directly on the precision and recall of high-severity, decision-relevant findings and assess whether severity-based filtering improves reviewer efficiency.

**Requested Changes:**

I would be willing to raise my score if the authors could address (partially or fully) the methodological deficiencies of the current framework. My main requests are summarized as follows:
- Conduct human evaluation on a substantially larger and more diverse set of papers, as opposed to just 5 in the current analysis. Ideally, and if possible, with more than one humans.
- Conduct exhaustive human audits of a manageable subset of codebases and compare AuditOwl’s outputs against that inventory. The current experiment measures self-rediscovery within the set of issues found by the model itself, rather than recall of all genuine issues.
- Use a genuinely independent verifier, such as a different model family or human verification, and report an ablation quantifying how much the verification stage improves factual and decision-relevant precision. The current same-family setup does not rule out shared systematic errors.
- The authors should either test the assumption that hallucinations are weakly correlated or weaken the conclusion that run-to-run variation mainly reflects missed true issues. The overall rediscovery rate is only 0.38, with substantially lower rates for medium- and low-severity findings.


A nice to have (but non-critical) would be to prioritize severe, reviewer-relevant findings. For instance, the authors could report precision and recall separately by severity and relevance. A reviewer-facing tool should demonstrate that it reduces rather than increases reviewer workload.